# A Novel Metabolite as a Hapten to Prepare Monoclonal Antibodies for Rapid Screening of Quinoxaline Drug Residues

**DOI:** 10.3390/foods11203305

**Published:** 2022-10-21

**Authors:** Wanyao Song, Mengyu Luo, Huaming Li, Jiaxu Xiao, Xiuping He, Jixiang Liang, Dapeng Peng

**Affiliations:** 1National Reference Laboratory of Veterinary Drug Residues (HZAU), MOA Key Laboratory for the Detection of Veterinary Drug Residues in Foods, Huazhong Agricultural University, Wuhan 430070, China; 2Shenzhen Institute of Nutrition and Health, Huazhong Agricultural University, Shenzhen 518000, China; 3Shenzhen Branch, Guangdong Laboratory for Lingnan Modern Agriculture, Agricultural Genomics Institute at Shenzhen, Chinese Academy of Agricultural Sciences, Shenzhen 518000, China; 4Genome Analysis Laboratory of the Ministry of Agriculture, Agricultural Genomics Institute at Shenzhen, Chinese Academy of Agricultural Sciences, Shenzhen 518000, China

**Keywords:** desoxymequindox, monoclonal antibody, ELISA, residue analysis

## Abstract

Quinoxalines (Qx) are chemically synthesized antibacterial drugs with strong antibacterial and growth-promoting effects. Qx is heavily abused by farmers, resulting in large residues in animal-derived foods, which pose a serious threat to human health. Desoxyquinoxalines (DQx), which have the highest residue levels, have been identified as the major toxicant and have become a new generation of residue markers. In this study, we prepared monoclonal antibodies (mAb) based on a new generation metabolite (desoxymequindox, DMEQ) and establish an indirect competitive enzyme-linked immunosorbent assay (ic-ELISA) for the rapid determination of Qx residues in food. The mAb exhibited high sensitivity with half maximal inhibitory concentration (IC_50_) and a linear range of 2.84 µg/L and 0.8–12.8 µg/L, respectively. Additionally, the cross-reactivity (CR) of the mAb showed that it recognized multiple DQx to varying levels. The limits of detection (LOD), limits of quantification (LOQ), and recoveries for the ic-ELISA assay of pork, swine liver, swine kidney, chicken, and chicken liver were 0.48–0.58 µg/kg, 0.61–0.90 µg/kg, and 73.7–107.8%, respectively, and the coefficients of variation (CV) were less than 11%. The results of the ic-ELISA showed a good correlation with LC–MS/MS in animal-derived foods. This suggests that this analytical method can be used for the rapid screening of QX residues.

## 1. Introduction

Quinoxalines (Qx) are chemically synthesized antibacterial drugs that have a strong inhibitory effect on both gram-positive and gram-negative bacteria [1,2]. Due to the broad antimicrobial spectrum and obvious growth-promoting effects, Qx are used in excessive amounts by farmers as feed additives to obtain more economic benefits [3]. However, several studies have shown that Qx can cause toxic side effects such as trichothecation, photosensitivity, and adrenal cortical damage in livestock and poultry. More seriously, drug residues in animal-derived foods can pose a significant threat to the population directly consuming them [4,5,6]. According to the latest reports, the quinoxalines desoxymetabolites (desoxyquinoxalines, DQx) have the highest residues and have been identified as major toxicants. They will soon replace 3-methyl-quinoxaline-2-carboxylic acid (MQCA) and quinoxaline-2-carboxylic acid (QCA) as the new residue markers for Qx [1,7,8]. The chemical structures of Qx and the major DQx are shown in Figure 1. The Ministry of Agriculture and Rural Affairs of the People’s Republic of China has stated that olaquindox (OLA), carbadox (CBX), and mequindox (MEQ) have been banned for use in food animals, and the requirements for residue detection of quinoxalines have been increasing. In order to accurately monitor the residual hazards of Qx in animals and reduce the risk to human health, there is a need to develop a residue-monitoring method for DQx.

Currently, many detection assays have been reported but mainly for Qx prototypes or MQCA and QCA, which were previously identified as residue markers. Additionally, the detection of QX is mainly focused on instrumental detection assays, such as high-performance liquid chromatography (HPLC) [9,10] and liquid-chromatography tandem mass spectrometry (LC–MS/MS) [11,12]. Though the instrument-detection method has high sensitivity and precision, it requires rather expensive instruments, complicated technology, a long operation time, and low detection efficiency. In contrast, immunoassay methods, such as enzyme-linked immunosorbent assay (ELISA), are simple, convenient, sensitive, specific and inexpensive. ELISA has become an alternative analytical method or a meaningful complement, especially in the detection of large numbers of samples [13,14]. Current indirect competitive ELISAs (ic-ELISA) are all for Qx prototypes, MQCA, or QCA [15,16]. To the best of our knowledge, immunoassay assays for the residue detection of DQx have not been previously reported. The maximum residue limits (MRLs) for DQx are not legally standardized, leading to potential gaps in food safety. Therefore, it is important to develop a rapid and accurate detection method for DQx.

In this study, we propose to synthesize novel haptens against desoxymequindox (DMEQ) to fully expose the main part of DQx. The aim is to prepare monoclonal antibodies (mAb) against DQx and establish an ic-ELISA method to provide strong technical support for ensuring the food safety of animal origin and reducing the risk to human health.

## 2. Materials and Methods

### 2.1. Chemicals and Apparatus

Quinocetone (QCT), OLA, MEQ, CBX, cyadox (CYX), DMEQ, desoxyolaquindox (DOLA), desoxyquinocatone (DQCT), desoxycarbadox (DCBX), desoxycyadox (DCYX), N1-desoxycyadox (N1-DCYX), N4-desoxycyadox (N4-DCYX), MQCA, and QCA were purchased from Sigma (USA). N-acetylsulfanilyl chloride, p-aminobenzoic acid (PABA), carboxymethyl hydroxylamine (AOAA), N-hydroxysuccinimide, pyridine, bovine serumalbumin (BSA), human serum albumin (HSA), dimethyl sulfoxide (DMSO), ovalbumin (OVA), polyethylene glycol (PEG), hypoxanthine–aminopterin–thymidine (HAT) medium, hypoxanthine–thymidine (HT) medium, and peroxidase-labelled goat anti-mouse immunoglobulins (HRP-IgG) were purchased from Sigma-Aldrich (USA). Australian fetal bovine serum was purchased from Thermo Fisher Bioengineering Materials (Beijing, China). The SP2/0 mouse-tumor cell line was obtained from our laboratory. All other chemicals and organic solvents were of analytical grade or better. Female Balb/c mice (6–8 weeks old, NO. 42000600000485) were bought from Three Gorges University (Yichang, China) and quality-tested by the Hubei Provincial Center for Disease Control and Prevention (Wuhan, China).

A full-wavelength microplate reader (BioTek, Winooski, VT, USA) and a UV spectrophotometer (Model 8453) were purchased from Agilent Technologies, Inc. (Santa Clara, CA, USA).

### 2.2. Antigen Design and Preparation

#### 2.2.1. Synthesis of Antigens MQCA–PABA–BSA/OVA

The synthesis protocol of MQCA–PABA–BSA/OVA is shown in Figure 2A. Briefly, MQCA (90.0 mg), dicyclohexylcarbodiimide (DCC, 110.0 mg), and N-hydroxysuccinimide (NHS, 55.0 mg) were dissolved with 1.2 mL of N, N-dimethylformamide (DMF), and the mixture was stirred at 4 °C overnight. After removing the precipitate by centrifugation at 6000× *g*, the supernatant was obtained, PABA (68.0 mg) was added, and the reaction was continued for 24 h with stirring. After centrifugation, the supernatant was collected and the solution became turbid after adding distilled water (10 mL). The precipitate obtained by filtering through filter paper is MQCA–PABA.

Then, MQCA–PABA (30.0 mg), NHS (15.0 mg), and DCC (20.2 mg) were dissolved with 1.0 mL of DMF, and the mixture was stirred at 4 °C overnight. After centrifugation, the supernatant was added slowly with precipitate to the solution of 15 mL phosphate-buffered saline (PBS, 0.01 mol/L, pH 7.4) containing 66.0 mg BSA. The mixture was stirred for 24 h at 4 °C and then dialyzed in PBS for 5 d, and the dialysate was changed every 8 h. The dialyzed sample (MQCA–PABA–BSA) was collected and stored at −20 °C until use. The same principle can be followed to obtain MQCA–PABA–OVA.

#### 2.2.2. Synthesis of Antigens DMEQ–AOAA–HSA/OVA

As shown in Figure 2B, AOAA (130.0 mg) was added to 6 mL of pyridine containing DMEQ (200.0 mg) and reacted at 60 °C for 6–8 h. Then, the pyridine was removed with a rotary evaporator, and 20 mL of saturated sodium bicarbonate solution and 50 mL of ethyl acetate were added. After shaking well, the aqueous layer was adjusted to pH 3 with hydrochloric acid and extracted twice with ethyl acetate. Finally, it was dried with anhydrous magnesium sulfate, and the filtrate was evaporated after filtration to obtain the solid: DMEQ–AOAA. The hapten was conjugated to the carrier protein in the same way as above, except that BSA was replaced with HSA. The haptens, carrier proteins, and conjugates were scanned under a UV spectrophotometer (200–400 nm) to determine the state of conjugation and to record the corresponding maximum absorption wavelength and absorption values. The relevant absorbance coefficient (K) was calculated from K = A/CL, and the conjugation ratio was calculated from [K(conjugation) − K(carrier protein)]/K(hapten).

### 2.3. Preparation of mAb

The preparation of mAb was performed according to that described in our laboratory by Peng et al. [17]. Briefly, sixteen female Balb/C mice were randomly divided into four groups for culture. Two immunogens (MQCA–PABA–BSA and DMEQ–AOAA–HSA) were immunized at a dose of 50 and 100 µg each. Then, serum titers and specificity were monitored by conventional ELISA procedures using MQCA–PABA–OVA and DMEQ–AOAA–OVA as homologous and heterologous coating antigens, respectively [18]. The mice with the best titer and specificity were selected to obtain splenocytes in vitro and fused with mouse myeloma cells. Under screening by the ELISA method, positive monoclonal cell lines were obtained after subcloning the fused cells using the limited dilution method 3–5 times. Finally, mAb was prepared by ascites induction, purified using protein A affinity chromatography, and stored at −20 °C until use.

### 2.4. Development of ic-ELISA Analysis

First, the purified antibody was titrated with different coating antigens in a square matrix to find the best concentration so that the sensitivity of the ic-ELISA assay would be the highest [19]. Then, the gradient standard solutions of DMEQ (0.8, 1.6, 3.2, 6.4, and 12.8 μg/L) were analyzed by ic-ELISA using the optimal concentrations optimized as above. The absorbance values obtained (OD_450nm_) were plotted against the corresponding concentrations in a standard calibration curve, and the concentration of DMEQ that produced half inhibition (IC_50_ value) was obtained from the results.

The extent of cross-reactivity (CR) of this mAb to different drugs was calculated by measuring the IC_50_ values of different structural analogs. Five DQx (DMEQ, DOLA, DQCT, DCYX, and DCBX) and six structural analogs (N1-DCYX, N4-DCYX, MQCA, MQCA–PABA, MEQ, and QCA) were used to determine the CR of the mAb. The IC_50_ values of all compounds were determined from the measured OD_450nm_ with the standard curve, and based on the IC_50_ values, the CR was calculated as follows: CR (%) = [IC_50_ (standard drug)/IC_50_ (analogues)] × 100%

### 2.5. Validation of ic-ELISA Analysis

Tissue samples of pork, swine liver, swine kidney, chicken, and chicken liver were purchased from local supermarkets. The homogenized samples (1 ± 0.05 g) were weighed, and 12 mL of extractant (ethyl acetate: acetonitrile = 1:1) was added to them. The mixture was mixed thoroughly for 3 min and then centrifuged at 4000× *g* for 10 min at room temperature. The organic phase was pipetted, and 2 mL of the aqueous NaOH solution (0.5 mol/L) was added. After shaking for 5 min, all organic phases were blown dry under nitrogen at 40 °C in a water bath. Then, the residue was resuspended with 1 mL of the compound solution (PBS: methanol = 9:1) and diluted 16 times with PBS to be ready for ic-ELISA analysis.

The sensitivity of the ic-ELISA analysis for practical applications is determined by the detection of the actual samples. Twenty different blank samples (pork, swine liver, swine kidney, chicken, and chicken liver) that were confirmed to be free of drug residues by LC–MS/MS were performed to ic-ELISA analysis. The mean (C) of the concentrations and the standard deviation (SD) were calculated for all blank samples. The limit of detection (LOD) and limit of quantification (LOQ) of this ic-ELISA method were calculated according to LOD = C + 3 × SD and LOQ = C + 10 × SD.

To demonstrate the accuracy and precision of the ic-ELISA analysis, the recovery rate and coefficient of variation (CV) were evaluated as indicators. DMEQ standard solutions were added to different types of homogenized samples at final concentrations of 1 × LOQ, 2 × LOQ, and 4 × LOQ, with 5 replicates of each concentration. The samples were mixed well by vortexing and left to stand for 1 h to allow full absorption of the drug, followed by sample pretreatment and ic-ELSIA analysis. The calculating equation is as follows: recovery = (actual concentration/spiked concentration) × 100%. CV = (SD/C) × 100%

To confirm the reliability of the developed ic-ELISA method, four spiked chicken samples were selected for comparison using ic-ELISA analysis and LC–MS/MS. The detailed procedures for LC–MS/MS analysis and sample preparation were performed according to the previous description [20]. Briefly, 10 mL of methanol-water (*v*/*v*, 5:95) and 20 mg of N-propyl ethylenediamine were added to the homogenized samples. They were vortexed for 1 min and then centrifuged at 15,000× *g* for 10 min, and the supernatant was collected and filtered through a 0.22 μm membrane. A Thermo Hypersil Gold column (150 × 2.1 mm i.d., 5.0 μm) was used for LC–MS/MS analysis, and water-formic acid (*v*/*v*, 100:0.1) and acetonitrile were used for the mobile phases. For the analysis of spiked chicken samples by ic-ELISA and LC–MS/MS, correlation fitting curves were drawn to assess the correlation between the two methods.

## 3. Results and Discussion

### 3.1. Antigen Design and Characterization

In this study, quinoxaline haptens were designed with MQCA and DMEQ as the precursor substances and PABA and AOAA as the arms, respectively. The chemical structure of the precursor substances fully exposes the common structure of DQx and also carries reactive groups to facilitate the synthesis of hapten. Thus, the mAb prepared from such an immunogen could maximize the simultaneous recognition of multiple DQx. The haptens of MQCA–PABA and DMEQ–AOAA are small molecules that do not induce an immune response in the body and need to be conjugated with large proteins to be immunogenic. The prepared hapten was used as the basis for the synthesis of the conjugated product by the NHS active ester method. With the free carboxylic group, MQCA–PABA and DMEQ–AOAA were individually linked to proteins [21,22]. BSA and OVA, which served as the most common candidates of carrier protein, were adopted for the assay.

The spectra of the hapten, carrier protein, and conjugates were scanned using a UV–Vis spectrophotometer. The ultraviolet absorbance spectra of MQCA–PABA–BSA (λmax, 279 nm, 324 nm), MQCA–PABA–OVA (λmax, 280 nm, 322 nm), DMEQ–AOAA–HSA (λmax, 321 nm), and DMEQ–AOAA–OVA (λmax, 281 nm) were different from the carrier protein of the BSA (λmax, 279 nm) and OVA (λmax, 279 nm), and from the haptens of MQCA–PABA (λmax, 322 nm) and DMEQ–AOAA (λmax, 321 nm), which confirmed that the antigens were successfully synthesized (Figure 3). The estimated conjugation ratios of MQCA–PABA–BSA, MQCA–PABA–OVA, DMEQ–AOAA–HSA, and DMEQ–AOAA–OVA were 16.9, 10.7,17.8, and 11.3, respectively, obtained from the absorbance calculation analysis.

### 3.2. Characterization of mAb

The immunogens MQCA–PABA–BSA and DMEQ–AOAA–HSA were inoculated into mice to produce mAb. After the third injection, the specificity and titer of the immunized mouse antisera were determined using homologous and heterologous coatings, respectively (Table 1). The analysis in the table shows that DMEQ–AOAA–HSA showed poor specificity as an immunogen, both in homologous and heterologous coating. The immunogen MQCA–PABA–BSA with the coating antigen MQCA–PABA–OVA showed high titers but poor specificity against DMEQ. However, DMEQ–AOAA–OVA showed a better inhibition rate as a coating antigen for heterologous analysis. Therefore, splenocytes from mice immunized with MQCA–PABA–BSA were fused with myeloma cells. In the process of preparing an antibody, we similarly found that the heterologous coating antigen DMEQ–AOAA–OVA was more suitable for ic-ELISA than the homologous coating antigen MQCA–PABA–OVA. After multiple cell subcloning, a single strain of hybridoma cells 1A3 showed better recognition of DMEQ. Therefore, this cell line was selected for the production of monoclonal antibodies.

First, the obtained antibodies were optimized for the ic-ELISA conditions. The results showed that the ratio of 1:2000 mAb and 6.0 µg/mL of the coating antigen (DMEQ–AOAA–OVA) worked best, and the rest of the experiments were performed under this condition. As shown in Figure 4, the standard curve was fitted with the logarithmic value of DMEQ concentration as the *X*-axis and B/B0 as the *Y*-axis. The standard curve ranged from 0.8–12.8 µg/L, and the IC_50_ value was 2.84 µg/L based on the DMEQ standard solution. In Table 2, it can be seen that the CR of mAb showed varying levels of cross-reactivity to MQCA–PABA (44%), DOLA (27%), MQCA (2%), DQCT (1.3%), and QCA (1%) but showed no measurable CR (CR < 0.1%) with DCBX, DCYX, N1-DCYX, N4-DCYX, and MEQ.

### 3.3. Performance of the ic-ELISA Analysis

Simple sample pretreatment will directly affect the analytical results of the ELISA. A large number of compounds in the sample matrix often have a strong interfering effect on the assay [19]. In previous studies, ethyl acetate was added to animal tissues for extraction and needed to be incubated for 2 h. After nitrogen blowing, it was degreased with hexane and diluted with PBS. Other studies showed that samples can be extracted by sodium hydroxide and acetonitrile, and diluted with diluent (methanol: PBS = 5:95) [23,24]. In this study, sodium hydroxide, ethyl acetate and acetonitrile were selected for the rapid extraction of DMEQ from the actual samples. This extraction method is faster and more efficient than previous methods.

To characterize the detection performance of the assay, we measured the LODs, LOQs, accuracy, and precision of the assay. Table 3 shows the results of 20 different blank samples, with the LODs and LOQs of samples ranging from 0.47–0.58 µg/kg and 0.61–0.90 µg/kg, respectively. The recoveries of the above samples spiked with DMEQ at the levels of 1 × LOQ, 2 × LOQ, and 4 × LOQ are listed in Table 3, which were in the range of 73.7% to 107.8%. The CVs were less than 10.8%, and we can find that the swine and chicken livers have the highest CVs. This is due to the different metabolic pathways of MEQ in different species and the differences in the enzymes involved [25,26]. All of the above results indicate that the analytical method developed in this study is highly sensitive, with good sample pretreatment and a low coefficient of variation.

In order to determine the stability of this method, a study was investigated to correlate the stability of the coating antigen, the antibodies, and the DMEQ standard used in the assay system. The coated enzyme plates and the prepared antibodies were placed in a 37 °C thermostat and taken out for ELISA validation on days 0, 2, 4, 6, and 8, respectively. The stability of the method was evaluated using the titer and the IC_50_ value of the analysis as evaluation criteria. The results in Figure 5A,B show that the titer of the enzyme plate and the antibody remained above 80%, and the IC_50_ fluctuated within 20% after 8 d of storage at 37 °C. According to the empirical Arrhenius equation, this implies that the enzyme plates and antibodies can be stored for at least 12 months at 4 °C. We also measured the IC_50_ of the standard concentration of DMEQ at 4 °C every month. The results showed that the standard solution of DMEQ can still be stored at 4 °C for at least 6 months and keep the IC_50_ stable (Figure 5C).

In addition, the reliability of the method was verified by analyzing the four spiked chicken samples (0.5, 1.0, 2.0, and 4.0 µg/kg) by ic-ELISA and LC–MS/MS, respectively. Figure 6 shows the fitted correlation coefficient (R^2^) of 0.9973 between this analytical method and the instrumental analysis results in chicken. This demonstrates the reliability of the established ic-ELISA method for DMEQ, which provides strong technical support for the monitoring of Qx drug residues in food.

## 4. Conclusions

In this study, we creatively synthesized a novel hapten to generate mAb against DQx. Based on the mAb, we firstly developed an ic-ELISA method to detect the residues of Qx in animal-derived foods, which was rapid, accurate, and sensitive. The simple analytical method reduces the sample pretreatment time, ensures higher efficiency, and meets the requirements for Qx residue analysis. The prepared mAb and the developed ic-ELISA method can monitor the residues of Qx in animal-derived foods to ensure food safety and human health.

## Figures and Tables

**Figure 1 foods-11-03305-f001:**
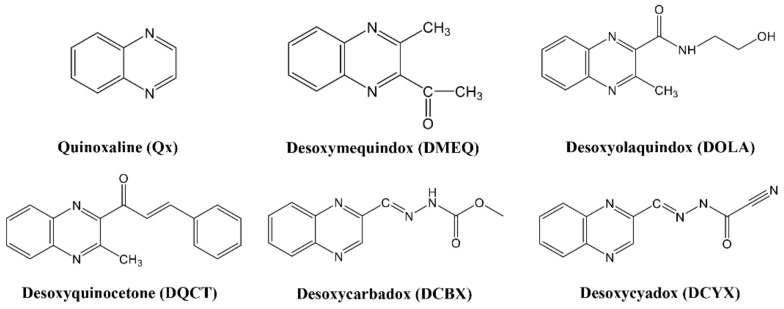
The chemical structures of Qx and the major DQx.

**Figure 2 foods-11-03305-f002:**
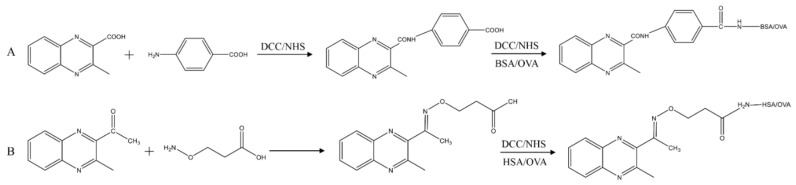
The synthesis protocol of MQCA–PABA–BSA/OVA (**A**) and DMEQ–AOAA–HSA/OVA (**B**).

**Figure 3 foods-11-03305-f003:**
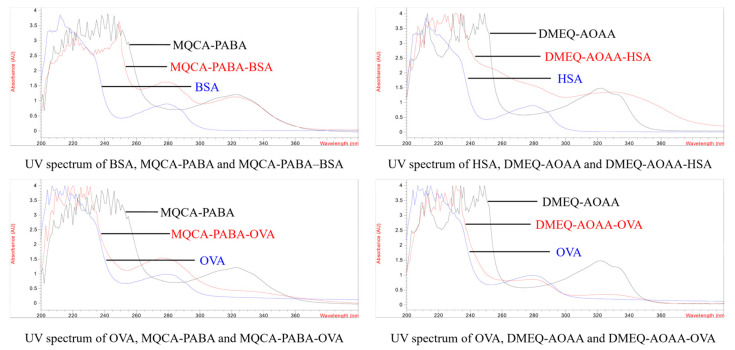
UV absorption peak spectra of 4 different antigens at 200–400 nm.

**Figure 4 foods-11-03305-f004:**
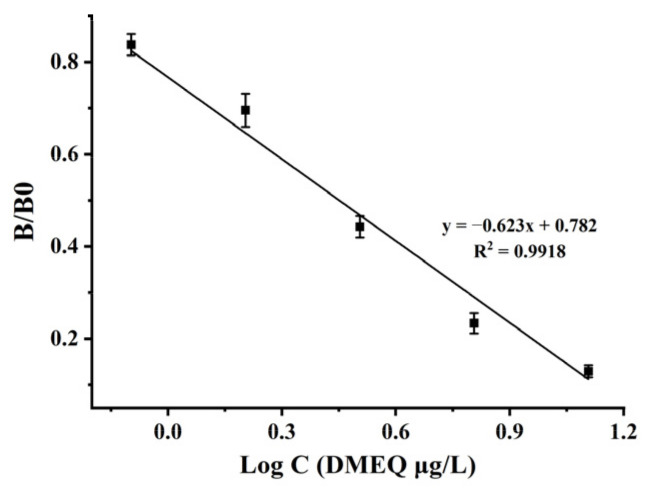
The standard curve of ic-ELISA.

**Figure 5 foods-11-03305-f005:**
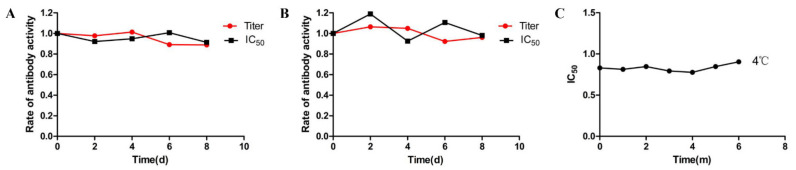
Stability testing of DMEQ–AOAA–OVA coating antigen (**A**), antibody (**B**), and DMEQ standard solution (**C**).

**Figure 6 foods-11-03305-f006:**
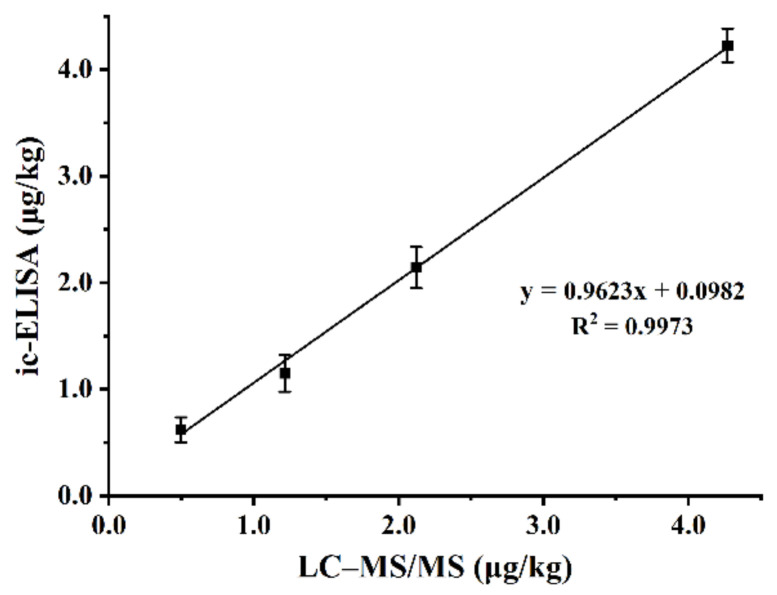
Correlation of LC–MS/MS and ic-ELISA for the analysis of spiked chicken samples.

**Table 1 foods-11-03305-t001:** The titer and specificity of the antiserum.

Immunogen	Coating Antigen	Titre (1:X × 10^3^)				B/B0 ^1^ Values (DMEQ, 100 μg/L)			
		Mouse 1	Mouse 2	Mouse 3	Mouse 4	Mouse 1	Mouse 2	Mouse 3	Mouse 4
MQCA–PABA–BSA	MQCA–PABA–OVA	2	1.5	12	3.5	0.846	0.965	0.815	0.902
	DMEQ–AOAA–OVA	1	0.8	2	1.5	0.516	0.469	0.568	0.766
DMEQ–AOAA–HSA	MQCA–PABA–OVA	2	3	1.8	1.5	0.851	0.921	0.956	0.827
	DMEQ–AOAA–OVA	3.5	5	4	3	0.790	0.873	0.884	0.919

^1^ B is the absorbance value of DMEQ addition, and B0 is the absorbance value of PBS addition of equal amount.

**Table 2 foods-11-03305-t002:** The cross-reactivity of mAb to different analytes.

Competitor	IC_50_ (µg/L)	CR (%)
DMEQ	2.84	100
MQCA–PABA	6.45	44
DOLA	10.52	27
MQCA	142	2
DQCT	218	1.3
QCA	284	1
DCBX	>1000	<0.1
DCYX	>1000	<0.1
N1-DCYX	>1000	<0.1
N4-DCYX	>1000	<0.1
MEQ	>1000	<0.1

**Table 3 foods-11-03305-t003:** LOD, LOQ, and CVs of the samples spiked with DMEQ.

Sample	LOD (µg/kg)	LOQ (µg/kg)	Spiked Level (μg/kg)	Recovery (%)	CV_intra-assay_ (%, *n* ^1^ = 3)	Mean Recovery ± SD (%)	CV_inter-assay_ (%, *n* ^1^ = 9)
Pork	0.47	0.61	0.6	93.3–95.7	<10.8	94.7 ± 1.2	1.3
			1.2	87.5–101.8	<9.7	94.3 ± 7.2	7.6
			2.4	86.3–89.8	<9.3	88.6 ± 2.0	2.3
Swine liver	0.58	0.90	0.9	89.6–104.0	<7.6	99.0 ± 8.2	8.2
			1.8	89.0–97.9	<10.1	94.4 ± 4.7	5.0
			3.6	95.9–107.8	<4.5	100.6 ± 6.3	6.2
Swine kidney	0.55	0.77	0.75	80.8–86.1	<8.9	82.9 ± 5.7	6.9
			1.5	79.3–94.8	<8.8	85.6 ± 8.8	10.3
			3.0	73.7–80.8	<6.0	78.2 ± 4.5	5.8
Chicken	0.52	0.74	0.75	82.4–99.5	<8.3	89.2 ± 9.1	10.2
			1.5	93.7–99.2	<6.6	96.0 ± 2.9	3.0
			3.0	81.3–89.5	<7.8	85.1 ± 4.1	4.8
Chicken liver	0.54	0.77	0.75	93.9–99.7	<5.1	97.2 ± 4.6	4.7
			1.5	76.4–82.9	<2.7	79.3 ± 3.2	4.0
			3.0	91.2–101.1	<8.2	97.0 ± 7.5	7.7

^1^ “*n*” means the number of parallel detection.

## Data Availability

Data is contained within the article.

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
