# Peer review of "A Novel Metabolite as a Hapten to Prepare Monoclonal Antibodies for Rapid Screening of Quinoxaline Drug Residues"

_foods, 2022, doi:10.3390/foods11203305_

Round 1
Reviewer 1 Report
In this manuscript, the authors established a competitive ELISA for the rapid screening of antibiotics in edible animal tissues. The overall quality of this manuscript is acceptable. However, the methodology portion should include more details, such as the sample preparation and stability determination. There are some grammar mistakes throughout the manuscript, which need to be revised. Please find my other comments below.
L26-27: Rewrite this sentence. The word “respectively” refers to LOD, LOQ and recoveries instead of pork, swine liver, and kidney.
L64-65: Current icELISA are all for Qx prototypes.
L80: Please keep it consistent whether to include the location of the supplier.
L104: Overnight.
L107: It is a little bit confusing here. Based on my understanding, after adding water and filtration, the precipitates on filter paper were collected as MQCA-PABA.
L120: pH 3.
L158-159: Missing subjective in this sentence.
L165: What are those blank samples?
L175: How to prepare the spiked chicken samples?
L198: How to perform the calculation?
Figure 3: I suggest the authors change the corresponding label/line into red, blue and black colors to make them easier to differentiate.
L204: What are homologous and heterologous coatings? Please explain.
For the standard curve in Table 2, please include the standard deviation information.
L264-265: Please rewrite this sentence. It is not very clear.
L266-267: How to convert the storage stability from 37 °C to 4 °C?
Table 3: Since in the standard curve, the interpolated value would be µg/L, how to convert this unit to µg/kg? In addition, please specify the working range of this assay.
Author Response
Reviewer 1:
In this manuscript, the authors established a competitive ELISA for the rapid screening of antibiotics in edible animal tissues. The overall quality of this manuscript is acceptable. However, the methodology portion should include more details, such as the sample preparation and stability determination. There are some grammar mistakes throughout the manuscript, which need to be revised. Please find my other comments below.
Response: Thank you very much for your encouraging and helpful comments regarding our manuscript.
- L26-27: Rewrite this sentence. The word “respectively” refers to LOD, LOQ and recoveries instead of pork, swine liver, and kidney.
Response: Thank you very much for pointing out the shortcomings of our manuscript. The word "respectively" in this sentence is indeed ambiguous, and we have rewritten it and highlighted it in red in the manuscript.
- L64-65: Current icELISA are all for Qx prototypes.
Response: Thank you very much for pointing out the shortcomings of our manuscript. The sentence has been simplified and modified according to your comments and it is highlighted in red in the manuscript.
- L80: Please keep it consistent whether to include the location of the supplier.
Response: Thank you very much for pointing out the shortcomings of our manuscript. We have made additions to the location of the supplier and conducted a thorough check for consistency in "Materials". The additional changes have been highlighted in red in the manuscript.
- L104: Overnight.
Response: Thank you very much for pointing out the shortcomings of our manuscript. The wording of the sentence is indeed not standardized, we have revised it and marked it in red in the manuscript for your convenience.
- L107: It is a little bit confusing here. Based on my understanding, after adding water and filtration, the precipitates on filter paper were collected as MQCA-PABA.
Response: Thank you very much for your meaningful questions about our manuscript. Your understanding is correct. Turbidity occurs when water is added to the supernatant, and the precipitate obtained after filtering through filter paper is MQCA-PABA. We have also made a more detailed explanation and marked it in red in the manuscript.
- L120: pH 3.
Response: Thank you very much for pointing out the shortcomings of our manuscript. We have made revisions in the manuscript and marked it in red.
- L158-159: Missing subjective in this sentence.
Response: Thank you very much for pointing out the shortcomings of our manuscript. We have made revisions for this sentence in the manuscript and marked it in red.
- L165: What are those blank samples?
Response: Thank you very much for your meaningful questions about our manuscript. Blank samples were pork, swine liver, swine kidney, chicken and chicken liver purchased at local markets as described above. We took 20 samples of each type to determine the sensitivity of the method in actual samples. We have explained it clearly in the manuscript and made the corresponding revisions, thank you again.
- L175: How to prepare the spiked chicken samples?
Response: Thank you very much for your meaningful questions about our manuscript. We added the appropriate amount of DMEQ standards solutions to the homogenized samples to give a final concentration of 1×LOQ, 2×LOQ, and 4×LOQ. The samples were mixed well by vortexing and left to stand for 1 h to allow full absorption of the drug. The spiking samples were prepared as described above, followed by sample pretreatment operations and ic-ELSIA analysis. We have also explained in detail in the manuscript and highlighted the changes in red for your review, thank you again.
- L198: How to perform the calculation?
Response: Thank you very much for your meaningful questions about our manuscript. The absorbance coefficients of the hapten and the carrier protein at their respective maximum absorption wavelengths were calculated according to the formula K=A/CL. Then the conjugation ratios were calculated from [K(conjugation)-K(carrier protein)]/K(hapten). We have also explained the calculations in detail and highlighted them in red on lines 128-130 of the manuscript, thank you again.
- Figure 3: I suggest the authors change the corresponding label/line into red, blue and black colors to make them easier to differentiate.
Response: Thank you for your very helpful suggestions for our manuscript. We have changed the corresponding label/line into red, blue and black colors in Figure 3. This really helps the reader to differentiate, thank you again.
- L204: What are homologous and heterologous coatings? Please explain.
Response: Thank you very much for your meaningful questions about our manuscript. Homologous and heterologous coatings are terms derived from the ELISA assay. As an example from this study, we used the monoclonal antibody produced by MQCA-PABA-BSA as an immunogen to perform the ic-ELISA analysis. If we use MQCA-PABA-OVA as the coating antigen for ic-ELISA analysis, the immunogen and the coating antigen have the same hapten only the carrier protein is different, so it is homologous coating. On the contrary, if we use DMEQ-AOAA-OVA as the coating antigen for the ic-ELISA analysis, the haptens of the immunogen and the coating antigen are different, so it is a heterologous coating. It has been shown that heterologous coating can sometimes significantly improve the sensitivity of ic-ELISA analysis, so this is an essential factor in the analysis. We hope you find our explanation clear and if this is of interest to you, we would be honored to talk to you in more details, thanks again.
- For the standard curve in Table 2, please include the standard deviation information.
Response: Thank you very much for pointing out the shortcomings of our manuscript. We did overlook the standard deviation information, which has now been refined and shown separately in Figure 4.
- L264-265: Please rewrite this sentence. It is not very clear.
Response: Thank you very much for pointing out the shortcomings of our manuscript. The sentence does appear to be somewhat unclear, and we have revised it and highlighted it in red in the manuscript.
- L266-267: How to convert the storage stability from 37 °C to 4 °C?
Response: Thank you very much for your meaningful questions about our manuscript. We are projecting the conversion based on the Arrhenius equation. According to the Arrhenius equation, it is generally considered that 1 d at 37°C is equivalent to 1.5 months at 4-10°C. In this study, the coating antigen and antibody were able to be stored at 37°C for 8 days, so they can be stored at 4°C for 12 months. We apologize for not explaining this clearly in the manuscript, and have now completed the changes and highlighted them in red. Thank you again.
- Table 3: Since in the standard curve, the interpolated value would be µg/L, how to convert this unit to µg/kg? In addition, please specify the working range of this assay.
Response: Thank you very much for your meaningful questions about our manuscript. In ic-ELISA analysis, standard curves are established based on different concentrations of standard solutions. The concentration of the standard solution is in µg/L, so the interpolated value would be µg/L in the standard curve. And the values in Table 3 are our measurements of the actual samples, which were calculated based on the standard curve. Since the test sample is a solid, this unit is converted to µg/kg. Moreover, µg/L and µg/kg belong to the same level of units and can be collectively referred to as ppb, only differing depends on the status of the tested sample. If the test sample is a liquid (e.g. milk), the unit of µg/L is more appropriate. The working range of the method has also been mentioned in the manuscript as 0.8-12.8 µg/L (Line 237-238).
Thank you again for your encouragement and helpful comments on our manuscript. We hope that all your questions have been clearly explained. If you still have any other questions, we would be happy to continue the communication and make further revisions to the manuscript.
Reviewer 2 Report
From the point of view of food safety, it is very important to screen the different residues from Qx in product of animal origin. A novel method is developed and validated to check the concentration of DQxs, which are the major toxic residues.
The manuscript is well edited, the results are clearly presented and show well-thought-out experimental work.
There are only some details that need to be completed or corrected.
1. Introduction
You write about the need of DQx measuring method, to replace the existing measurements for MQCA markers. Please explain in the manuscript why the synthesis of MQCA-PABA-BSA/OVA was also carried out.
2. Materials and Methods
Despite the indication of the reference, a short description of samples and sample preparation and also of the reference LC-MS/MS method is missing
2.3.It should be briefly mentioned why it uses a different synthesis route to produce the two types of antigen conjugates.
Table 2. The standard curve of ic-ELISA should be removed from the Table 2. and present as a figure. The standard deviation must be marked in the figure, the first point must not be less than 0.0 concentration. Check and correct the diagram.
Line 280. You should mention, that these results respond to spiked chicken samples.
In summary, the manuscript presents a novel and useful screening ic-ELISA immunoassay for food products of animal origin. The manuscript need to do minor revision.
Author Response
Review 2:
From the point of view of food safety, it is very important to screen the different residues from Qx in product of animal origin. A novel method is developed and validated to check the concentration of DQxs, which are the major toxic residues. The manuscript is well edited, the results are clearly presented and show well-thought-out experimental work. There are only some details that need to be completed or corrected.
Response: Thank you very much for your encouraging and helpful comments regarding our manuscript.
- Introduction
You write about the need of DQx measuring method, to replace the existing measurements for MQCA markers. Please explain in the manuscript why the synthesis of MQCA-PABA-BSA/OVA was also carried out.
Response: Thank you very much for your meaningful questions about our manuscript. This is based on the principle of hapten design for synthesis. DQx is a class of drugs, and if you want to prepare monoclonal antibodies that recognize more of the DQx drug, you need the immunogen to fully expose the shared structure of DQx. The precursor substances for the synthesis of hapten should also have reactive groups that can be easily synthesized chemically, such as -COOH and -NH2. The chemical structure of MQCA fully exposes the common structure of DQx (bicyclic structure) and also carries a reactive group (-COOH) to facilitate the synthesis of hapten. Therefore, we used MQCA as a precursor substance for the synthesis of hapten and performed experimental attempts, and good results showed that the idea is reliable. We have also included a more detailed explanation in the "Results" section of the manuscript and highlighted it in red. We hope you find our explanation clear and if this is of interest to you, we would be honored to talk to you in more details, thanks again.
- Materials and Methods
Despite the indication of the reference, a short description of samples and sample preparation and also of the reference LC-MS/MS method is missing
Response: Thank you very much for pointing out the shortcomings of our manuscript. This is indeed an oversight of our part. We have supplemented the short steps of sample preparation and LC-MS/MS analysis in detail (Line 186-191). The modified part has been marked in red for your checking, thank you again.
- It should be briefly mentioned why it uses a different synthesis route to produce the two types of antigen conjugates.
Response: Thank you very much for pointing out the shortcomings of our manuscript. Based on the principle of hapten design, this study wanted to prepare monoclonal antibodies that could recognize as many DQx as possible. This requires that the haptens in the antigen conjugates should maximize exposure of the common portion of DQx as an antigenic epitope for recognition by the immune system of mice. The synthetic routes of the two antigen conjugates designed in this study were simple and able to adequately expose the common part of DQx, and therefore it served as the experimental protocol. We have also made a short explanation in the manuscript and highlighted it in red (Line 195-199). Thank you again.
Table 2. The standard curve of ic-ELISA should be removed from the Table 2. and present as a figure. The standard deviation must be marked in the figure, the first point must not be less than 0.0 concentration. Check and correct the diagram.
Response: Thank you very much for pointing out the shortcomings of our manuscript. The standard curve for ic-ELISA has been removed from Table 2 and is shown separately in Figure 4. We did forget the standard deviation information and have now successfully added it. The standard curve is based on the logarithmic value of the concentration of the standard solution as the horizontal coordinate (X-axis). And the first concentration of the standard solution in this study was 0.8 µg/L (value <1). The logarithm of a value less than 1 is less than 0. Therefore, the horizontal coordinate of the first point of the standard curve is less than 0.0. Thank you very much for your careful review and thank you again.
Line 280. You should mention, that these results respond to spiked chicken samples.
Response: Thank you very much for pointing out the shortcomings of our manuscript. We have modified it accordingly and highlighted it in red in the manuscript.
In summary, the manuscript presents a novel and useful screening ic-ELISA immunoassay for food products of animal origin. The manuscript need to do minor revision.
Thank you again for your encouragement and helpful comments on our manuscript. We hope that all your questions have been clearly explained. If you still have any other questions, we would be happy to continue the communication and make further revisions to the manuscript.
Round 2
Reviewer 1 Report
The authors have addressed previous comments.